# Genetic analysis reveals an east-west divide within North American *Vitis* species that mirrors their resistance to Pierce's disease

**Summaira Riaz, Alan C. Tenscher, Claire C. Heinitz, Karla G. Huerta-Acosta, M. Andrew Walker** *

Department of Viticulture and Enology, University of California, Davis, California, United States of America

* awalker@ucdavis.edu

**Data Availability Statement:** All relevant data are within the manuscript and its Supporting Information files.

## Abstract

Pierce's disease (PD) caused by the bacterium *Xylella fastidiosa* is a deadly disease of grapevines. This study used 20 SSR markers to genotype 326 accessions of grape species collected from the southeastern and southwestern United States, Mexico and Costa Rica. Two hundred sixty-six of these accessions, and an additional 12 PD resistant hybrid cultivars developed from southeastern US grape species, were evaluated for PD resistance. Disease resistance was evaluated by quantifying the level of bacteria in stems and measuring PD symptoms on the canes and leaves. Both Bayesian clustering and principal coordinate analyses identified two groups with an east-west divide: group 1 consisted of grape species from the southeastern US and Mexico, and group 2 consisted of accessions collected from the southwestern US and Mexico. The Sierra Madre Oriental mountain range appeared to be a phylogeographic barrier. The state of Texas was identified as a potential hybridization zone. The hierarchal STRUCTURE analysis on each group showed clustering of unique grape species. An east-west divide was also observed for PD resistance. With the exception of *Vitis candicans* and *V. cinerea* accessions collected from Mexico, all other grape species as well as the resistant southeastern hybrid cultivars were susceptible to the disease. Southwestern US grape accessions from drier desert regions showed stronger resistance to the disease. Strong PD resistance was observed within three distinct genetic clusters of *V. arizonica* which is adapted to drier environments and hybridizes freely with other species across its wide range.

## Introduction

Pierce's Disease (PD), caused by the bacterium *Xylella fastidiosa*, is an important disease of European wine and table grapes (*Vitis vinifera* cvs.) and its strains cause disease in an expanding list of horticultural crops [1,2]. Pierce's disease limits grape cultivation in California [3], the southeastern US [4], Mexico [5], and Central America [6]. The history of PD goes back to the late 1800s with reports from across the coastal plain areas of Texas, Mississippi, Georgia, Alabama, Louisiana, Missouri, North and South Carolina, and California [3,7–10]. It was

**Funding:** MAW received funding for this study from the California Department of Food and Agriculture PD/GWSS Board (https://www.cdfa.ca.gov/pdcp/PD_GWSS_Board.html) and the Rossi Endowed Chair in Viticulture funds were also used. The funders had no role in study design, data collection and analysis, decision to publish, or preparation of the manuscript.

**Competing interests:** The authors have declared that no competing interests exist.

speculated that the failure of European grapes in Florida in 1850 was also due to PD [11] Early breeding programs in the southeastern US utilized wild *Vitis* species that did not show typical PD symptoms and could survive longer in the field [12,13]. The grape breeding programs in central Florida and Mississippi used local grape species, mostly accessions of *V. aestivalis* and *V. shuttleworthii*, to develop many PD resistant hybrids capable of surviving field trials [4,14–16]. Resistance to PD was also evaluated in *Vitis rotundifolia* [17,18], a grape species that has shown remarkable resistance to fungal diseases, a broad range of nematodes and grape phylloxera [19]. This taxon also possesses a broad array of genetic and morphological differences that distinguish it from all other *Vitis* species and creates sterile hybrids with them, leading to its inclusion in a separate subgenus or genus–*Muscadinia*.

Symptoms of PD infection are superficially similar to acute water stress. They include marginal leaf necrosis, leaf scorch, leaf blade drop leaving attached petioles, uneven cane lignification, shriveled fruit, die back and eventual death within one to five years [20]. Early PD-resistance breeding work lacks information about the pattern of bacterial infection and there are no reports that quantify bacterial levels in infected plants [14–16]. Pierce's disease resistance was primarily evaluated by the presence of visible PD symptoms in the field under natural infection. Wild accessions and breeding lines with reduced symptoms and that survived longer in field trials were assumed to be resistant. Based on field evaluations of breeding populations derived from three southeastern US grape species, it was postulated that resistance to PD in southeastern grape species is trigenic with resistance dominant to susceptibility [15]. The presence of the pathogen in southern states and early breeding work led to the hypothesis that the causal agent of PD in grapes, *X. fastidiosa* subsp. *fastidiosa*, originated in the southeastern US and that the native grape species from that region coevolved with the pathogen and developed resistance [3,14–16].

A challenge to this hypothesis emerged from the genomic analysis of *X. fastidiosa* isolates from different regions. Sequence analysis revealed that only *X. fastidiosa* subsp. *multiplex*, which does not cause disease in grapes, is native to the US [21]. Furthermore, subsp. *fastidiosa*, which causes disease in cultivated grapes, has limited genetic diversity within isolates collected from California, Texas and Florida indicating a possible recent introduction [22,23]. Comparisons of genome sequences suggested tropical Central America as the origin of subsp. *fastidiosa*, which separated from the subsp. *multiplex* (native to the eastern US) a minimum of 15,000 years ago and possibly more than 30,000 years ago [21,22]. Greater genetic diversity was observed in the isolates of subsp. *fastidiosa* from Costa Rica, and the isolates from the US were nested within that group. These results raised important questions about the origin of PD resistance in the southeastern US grape species given the absence of subsp. *fastidiosa*. From an evolutionary perspective, 150–200 years of exposure to the pathogen may not be enough time for grape species native to the eastern US to evolve resistance to this pathogen.

The 2006 discovery of strong PD resistance in grape species from northeastern Mexico [24], also cast doubt on the hypothesized southeastern US origin for subsp. *fastidiosa*. The resistant accession, b43-17, collected in Monterrey, Mexico, appears to be a hybrid of *V. arizonica* and *V. candicans* (syn. *V. mustangensis*) [25]. All plants in the F1 population from a cross of susceptible *V. rupestris* cv. A. de Serres and b43-17 were resistant to *X. fastidiosa*. A major resistance locus that segregated 1:1 in a pseudo-backcross population, was identified on chromosome (chr) 14 and it was named *PdR1* [24,26]. A recent study by Riaz et al. [27] identified nine genetically distinct PD resistant accessions collected from Mexico and the bordering states of Texas and Arizona. Three additional accessions (b41-13, b40-14 and T03-16) that are the subject of another study [28] were also found to have strong resistance to PD. The PD resistance of all 12 accessions genetically mapped to chr 14 at a similar genomic position to where the *PdR1* locus from b43-17 mapped. The widespread geographic distribution of a similar

resistance locus across different grape species in the southwestern US and Mexico raises questions about the origins of PD resistance in the southwest US and Mexico, how it differs from PD resistance in the southeastern US grape species, and what evolutionary forces shaped PD resistance in *Vitis* species from North, Central and South America.

The North American *Vitis* taxonomy and nomenclature is complex due to widespread hybridization among sympatric species and morphological variation within species [29,30]. In the recent 'Flora of North America' (http://www.efloras.org/florataxon.aspx?flora_id=1&taxon_id=134649), 19 grape species were recognized. However, it did not include Mexican grape species that have been described by Comeaux [31,32] and Comeaux and Lu [33], and that deserve special attention because Mexico exhibits great genetic diversity of flora and fauna within a complex transition zone where Nearctic and Neotropical biotas overlap [34,35]. The grape germplasm from this region reflects historical speciation events promoted by environmental and geological changes that occurred over thousands of years. The mountain system of the Sierra Madre in Mexico, the Rockies in US and the deserts of Sonora, and Chihuahua that cover both countries contain a rich biodiversity of grape species. *Vitis* species, with the exception of the muscadine species, are all interfertile, but remain distinct due to differences in their habitat preference, physical geographical barriers, and phenological differences in flowering dates. The desert regions of the southwestern US and Mexico create significant geographical barriers between isolated mountain ranges that provide unique niches for grape species where water is available. A better understanding of the phylogenetic relationships of grape germplasm from different geographical regions in the southwest and southeastern US and Mexico would help shed light on the origin and dispersal of PD resistance in the grape species collected from this important transition zone.

In this study, we used 20 Simple Sequence Repeat (SSR) markers to genotype 346 accessions representing 19 grape species collected from Costa Rica, Mexico, the southwestern states bordering Mexico (California, Arizona, New Mexico, and Texas), and the southeastern coastal states where PD greatly limits the cultivation of *V. vinifera* grapes. Two hundred and sixty-six accessions were phenotyped using an optimized greenhouse-based screening method (high temperature and water stress) to evaluate their resistance to *X. fastidiosa*. Bacterial amounts in the stem were quantified and PD symptoms (uneven cane lignification, leaf scorch and leaf loss) were recorded. Germplasm previously identified as PD resistant was also included in the study. Finally, allelic data from 20 SSR markers were used for genetic analysis. The first objective of the study was to evaluate the historical relationships among different grape species and to determine if there is a genetic continuum between eastern and western US *Vitis* germplasm. The second objective was to determine whether geographic and taxonomic associations exist for PD resistance. The final goal was to compare previously reported PD resistant breeding lines from the southeastern US to the southwestern US germplasm under similar screening conditions.

## Materials and methods

### Plant material

Tables 1 and S1 Table provide summary and detailed information, respectively, on the 346 accessions within 19 grape species. Species designations were made based on morphological characteristics of the field grown plants. All accessions reported in this study are maintained either by the Department of Viticulture and Enology, University of California, Davis, CA (UCD), or the National Clonal Germplasm Repository, USDA-ARS, Davis (NCGR-Davis), or both. Germplasm from Mexico was acquired as seeds or cuttings by H.P. Olmo in 1961, and later by B.L. Comeaux in 1991. US grape germplasm was collected as cuttings across the

**Table 1. Summary of evaluated germplasm by species.** Simple sequence repeat marker data were generated for 346 accessions from 19 grape species and 326 were included in the genetic analysis study. The greenhouse-based screen for PD resistance was completed for 266 accessions.

| Grape species | No. of accessions | No. included in genetic analysis | No. tested for Pierce's disease |
|---|---|---|---|
| *V. acerifolia* | 2 | 2 | 1 |
| *V. aestivalis* | 31 | 31 | 13 |
| *V. arizonica* | 118 | 116 | 110 |
| *V. berlandieri* | 22 | 18 | 20 |
| *V. biformis* | 7 | 7 | 0 |
| *V. champinii* | 4 | 4 | 4 |
| *V. cinerea* | 51 | 51 | 28 |
| *V. doaniana* | 5 | 5 | 4 |
| *V. girdiana* | 21 | 13 | 21 |
| *V. labrusca* | 5 | 5 | 3 |
| *V. monticola* | 8 | 8 | 7 |
| *V. candicans* | 12 | 10 | 12 |
| *V. nesbittiana* | 9 | 9 | 2 |
| *V. riparia* | 4 | 3 | 4 |
| *V. rupestris* | 11 | 8 | 11 |
| *V. shuttleworthii* | 4 | 4 | 4 |
| *V. tiliifolia* | 10 | 10 | 3 |
| *V. treleasei* | 18 | 18 | 17 |
| *V. vulpina* | 4 | 4 | 2 |
| Total | 346 | 326 | 266 |

southern states during collection trips from 1997 to 2016. Designated Davis *Vitis* Identification Tag (DVIT) names were used for accessions maintained by NCGR-Davis. Global positioning system (GPS) coordinates of the collection site are provided for recently collected material housed at UCD (S1 Table). For historic collections maintained at the NCGR-Davis, location coordinates are not available. In these cases, location was determined from the collection notes when possible.

## Disease evaluations

A total of 266 accessions were evaluated for PD resistance using the greenhouse-based screen described in earlier studies (Table 1) [27,36]. Data for an additional 80 accessions were not reported for various reasons: either they failed to propagate, had too few replicates or died during greenhouse screening. In cases where accessions were from a seed lot, three to five seedlings were selected for testing. A minimum of four biological replicates of each accession were tested. A total of 19 screening experiments were carried out from 2011 to 2018. Grape accessions with known strong and intermediate resistance to PD and the susceptible *V. vinifera* cultivar Chardonnay (un-inoculated and inoculated) were used as reference plant controls (hereafter called reference plants) in every screen. The use of these reference plants allowed us to compare screen results across different experiments and years.

Plants were propagated from hardwood or herbaceous cuttings taken from plants growing in the field at UCD or at the NCGR-Davis. Hardwood cuttings were soaked in water overnight, placed in a callusing media (60:40 ratio of perlite:vermiculite) and kept in a dark room at 100% relative humidity and 29°C for two weeks. After two weeks, cuttings were placed in 5 x 5 x 15cm paper sleeves with a 1:1 ratio of callus media:peat moss after trimming excess roots and dipping the exposed portion in wax to prevent water loss. The sleeved cuttings were kept for

an additional 3–4 days in the dark at conditions described above and were then transferred to beds in a fog room with 27˚C bottom heat for two weeks before transplanting to pots. Actively growing plants were propagated using two- to three-node herbaceous cuttings that were rooted in 2 x 2 x 6 cm cellulose plugs in a fog room with 27˚C bottom heat. Both herbaceous and hardwood rooted cuttings were transplanted to 1 L pots with 1:1:1 Yolo sandy loam soil/perlite/peat mix. To ensure uniform growth at the time of inoculation, after about 4 weeks of growth all plants were cut back to two buds and regrown. As the main shoots grew to 1 m all lateral shoots were removed routinely to promote better air circulation and light penetration. Plants were fertigated with a 25% Hoagland's solution (Sigma-Aldrich, St. Louis) via a drip system (130 ml daily per plant). Plants received both supplemental and ambient light for an average of 18 h per day.

A *X. fastidiosa* isolate collected from Yountville, Napa County, California was used for all screens. The bacteria were maintained in greenhouse grown susceptible Chardonnay plants and isolated using the procedures described by Krivanek et al. [37]. For inoculations, actively growing bacteria were washed from Petri plates with ddH$_2$O, and the cell suspension was standardized to a 0.25 absorbance at 600 nm (approximately $6 \times 10^8$ CFU/ml as determined by culture plating). Plants were needle inoculated [38] twice about 10–15 cm above the base of each shoot with a total of 20 μl of bacterial suspension. The plants were sampled to quantify the bacterial amount 10 to 14 weeks post inoculation when the susceptible reference plants started showing leaf scorch and uneven cane lignification. For each test plant, a 0.5 g section of stem tissue was taken 30 cm above the point of inoculation and placed into a grinding bag (Agdia, Elkhart, Indiana, USA) with 5 ml of phosphate-buffered saline (PBS), 0.05% Tween, and 2% soluble polyvinylpyrrolidone (PVP-40) buffer (Nome et al. 1981). Samples were lightly crushed with a hammer and further processed using a Homes 6 mechanical homogenizer (Bioreba, Longmont, Colorado, USA), and the resulting extract was stored at -20˚C before ELISA testing.

Disease severity was assessed by three different methods at 10–14 weeks post inoculation. The mean percentage area of leaf scorch and leaf loss (LS/LL) on the four leaves above and nearest to the point of inoculation (POI) were measured. The degree of cane maturation in terms of green islands and necrosis development designated as the cane maturation index (CMI) was measured as described in an earlier study [37]. Finally, ELISA was used to measure the *X. fastidiosa* levels in the stem [36]. To obtain homogeneous variances and normally distributed residuals, ELISA data were natural log transformed. All statistical analysis was performed using JMP Pro14 software (Copyright 2018, SAS Institute Inc.). The reference plant controls across the 19 screens were analyzed to determine the variability of ELISA values and a two-way ANOVA analysis was carried out with 'genotype', 'experiment take down (TK) date', and the interaction of the two factors. Least square means comparisons were made with Tukey's test based on a least significant difference (LSD) for the reference plants and 19 screens. The ELISA values, CMI, and LS/LL data of wild accessions was analyzed with the inclusion of the reference plants to adjust for variation among the screens.

## Genotyping

DNA was extracted from young leaf tissue using a modified CTAB protocol as described earlier with the exclusion of the RNase step [26]. Standard alcohol DNA precipitations were carried out following a single chloroform-isoamyl alcohol wash; DNA was dissolved in 1X TE buffer and stored at -20˚C for further use. A total of 24 SSR markers were used to develop fingerprint data (S2 Table). Amplifications for each marker were carried out separately. The PCR amplifications were performed in 10 μl reaction following the protocols described in an earlier

study [27]. Amplified products were combined depending on the amplicon size and fluorescent labels of the markers and run on an ABI 3500 capillary electrophoresis analyzer with GeneScan-500 Liz Size Standard (Life Technologies, Carlsbad, California, USA). Eight samples were used as an internal reference on each plate to standardize the allele calls between different runs on the ABI 3500. Allele sizes were determined using GeneMapper 4.1 software (Applied Biosystem Co., Ltd., USA).

## Genetic diversity analysis

STRUCTURE V2.3.1 was used to infer the number of clusters [39]. The algorithm was run for a range of genetic clusters (K) from 1 to 20 using the admixture model, and it was replicated 20 times for each K. Each run was implemented with a burn-in period of 100,000 steps followed by 100,000 Monte Carlo Markov Chain replicates using no prior information and assuming correlated allele frequencies. Structure Harvester [40] and CLUMPPAK [41] were used to process the STRUCTURE results. The optimum value of K was obtained by calculating the **Δk** value [42]. The bar plots were drawn with STRUCTURE PLOT (2.0) [43]. Simple matching distance and principal coordinate analysis (PCoA) was carried out with DARwin software (version 5.0.158) [44]. For the genetic diversity analysis, GenAlEx 6.5 software was used to calculate alleles observed (Na), observed heterozygosity ($H_O$), expected heterozygosity ($H_E$), coefficient of inbreeding ($F_{IS}$), genetic differentiation coefficient ($F_{ST}$), and gene flow (Nm) [45]. All maps showing STRUCTURE assignment, and PD evaluation results were created using ArcGIS® software by Esri (ArcGIS® and ArcMap™, www.esri.com) [46], using World topographic map as base layer (Sources: Esri, DeLorme, HERE, TomTom, Intermap, increment P Corp., GEBCO, USGS, FAO, NPS, NRCAN, GeoBase, IGN, Kadaster NL, Ordnance Survey, Esri Japan, METI, Esri China (Hong Kong), swisstopo, MapmyIndia, and the GIS User Community).

## Results

### Reliability of the greenhouse assay for disease evaluation

In this study, 19 screens from 2011 to 2018 were carried out to evaluate 266 accessions for PD resistance. To compare the results across multiple screens, seven accessions [two resistant (b43-17, U0505-01), three intermediate (U0505-35, Roucaneuf, Blanc du Bois), and two susceptible (U0505-22, Chardonnay)] were used as reference plants in every experiment. The S1 Fig shows the seven reference plants at 12 weeks post inoculation. The uninoculated Chardonnay control did not have measurable bacteria in 19 screens and was excluded from further analysis. The variability of greenhouse temperature during the year has an impact on disease expression and quantifiable bacterial levels (unpublished data). Therefore, 'genotype', 'experiment take down date', and the interaction of these two factors were statistically significant based on the two-way ANOVA (Table 2A). However, the reference plants were separated into three categories (resistant, intermediate and susceptible) in every screen (S2 Fig). In addition, least square means with Tukey's test separated the resistant, intermediate and susceptible reference plants when data from the 19 screens were combined (Table 2B). These results indicate that the greenhouse-based ELISA screen is a reliable method to distinguish PD resistant and susceptible accessions with reproducible results that could be combined and compared across multiple screens.

### Disease evaluations of diverse germplasm

The S1 Table presents the LSM of ELISA values, CMI, and LS/LL results for 266 accessions. Lower values in all three cases reflected better performance. For better display of the ELISA

**Table 2. a. Two-way ANOVA of the reference plants used as internal standards across 19 screens carried out from 2011 to 2018.** Experiment take down date is abbreviated as TD. Uninoculated Chardonnay was not included in the analysis. b. Least square means (LSM) comparisons of ELISA readings of bacterial levels using Tukey's test with take down date and controls as variables. Results significantly separate the resistant, intermediate and susceptible reference plants. Confidence level (CL) were established at 95%. Uninoculated Chardonnay was not included in the analysis.

| Genetic Background | No of parameters | Degree of Freedom | Sum of Squares | F Ratio | Prob > F |
|---|---|---|---|---|---|
| Reference Plant | 6 | 6 | 4661.54 | 366.72 | < .0001* |
| TD Date | 18 | 18 | 682.75 | 17.90 | < .0001* |
| Reference Plant * TD Date | 108 | 108 | 536.21 | 2.34 | < .0001* |
| **Genotype** | **LSM** | **Std. error** | **Lower CL** | **Upper CL** | **Group** |
| b43-17 | 9.02 | 0.15 | 8.73 | 9.31 | A |
| U0505-01 | 11.20 | 0.12 | 10.97 | 11.43 | B |
| U0505-35 | 13.25 | 0.12 | 13.02 | 13.48 | C |
| Blanc du Bois | 13.30 | 0.13 | 13.05 | 13.56 | CD |
| Roucaneuf | 13.81 | 0.13 | 13.55 | 14.07 | D |
| U0505-22 | 15.87 | 0.12 | 15.63 | 16.11 | E |
| Chardonnay | 16.30 | 0.12 | 16.06 | 16.54 | E |

values, CMI, and LS/LL results on the maps, five categories and a similar color scheme was used (Fig 1 and S3 Fig). The categories to display ELISA results were: 1 (6.9–9.1), 2 (9.2–11.1), 3 (11.2–13.1), 4 (13.2–15.1), and 5 (15.2 and above). The five categories for the CMI and LS/LL scores were: 1 (0–1), 2 (1.1–2.0), 3 (2.1–3.0), 4 (3.1–4), and 5 (4.1and above). In all three cases, category 1 and 2 defined resistant accessions, category 3 and 4 possessed intermediate to borderline susceptible accessions, and the susceptible accessions were in category 5. S3 Table presents the disease evaluation results in categories and accessions are organized from lower to higher values for ELISA, CMI, and LS/LL. Nineteen tested accessions were not included in the genetic analysis and therefore were not displayed on the maps in Fig 1 and S3 Fig. The correlation analysis indicates that overall ELISA results had a 75% and 68% correlation to CMI and LS/LL values, respectively, and CMI results had a 67% correlation with LS/LL.

Fig 1 and S3 Fig present the ELISA, CMI and LS/LL categories for each tested accession, respectively. For all three the same color scheme was used for the five categories (dark green = 1, light green = 2, yellow = 3, orange = 4, and burgundy = 5). Results indicate that there is an east-west divide for PD resistance formed by the Sierra Madre Oriental and central Texas. Most accessions from eastern Mexico, the bordering state of Texas and the southeastern US had higher values for all three resistance parameters in comparison to accessions from the southwestern US and Mexico. These regions have contrasting climates. Eastern Mexico and the Gulf Coast states are more tropical with higher humidity and rainfall in comparison to the southwestern US and northwestern Mexico, which have hotter and more arid conditions. Over all, the state of Texas possesses a rich *Vitis* flora with 14 species and two natural hybrids, and many sympatric zones where introgressive forms appear. The state also has a mixture of resistant, intermediate and susceptible accessions (S3 Table, Fig 1 and S3 Fig).

Fifty-two accessions grouped into category 1 with ELISA values ranging from 6.9–9.1 (S1 and S3 Tables). This group contained b43-17, ANU05, ANU71, b40-29, b46-43, SAZ7, and T03-16, all of which have PD resistance mapped to chr14. Forty-seven accessions in this group were called *V. arizonica* based on their morphological features and were collected from Mexico and the bordering states of Arizona, New Mexico, and Texas (S1 and S3 Tables). A majority of the accessions in this group had lower CMI and LS/LL scores than b43-17, with the exception of two accessions (AZ11-107, ANU71), which had better cane maturation but higher scores for LS/LL.

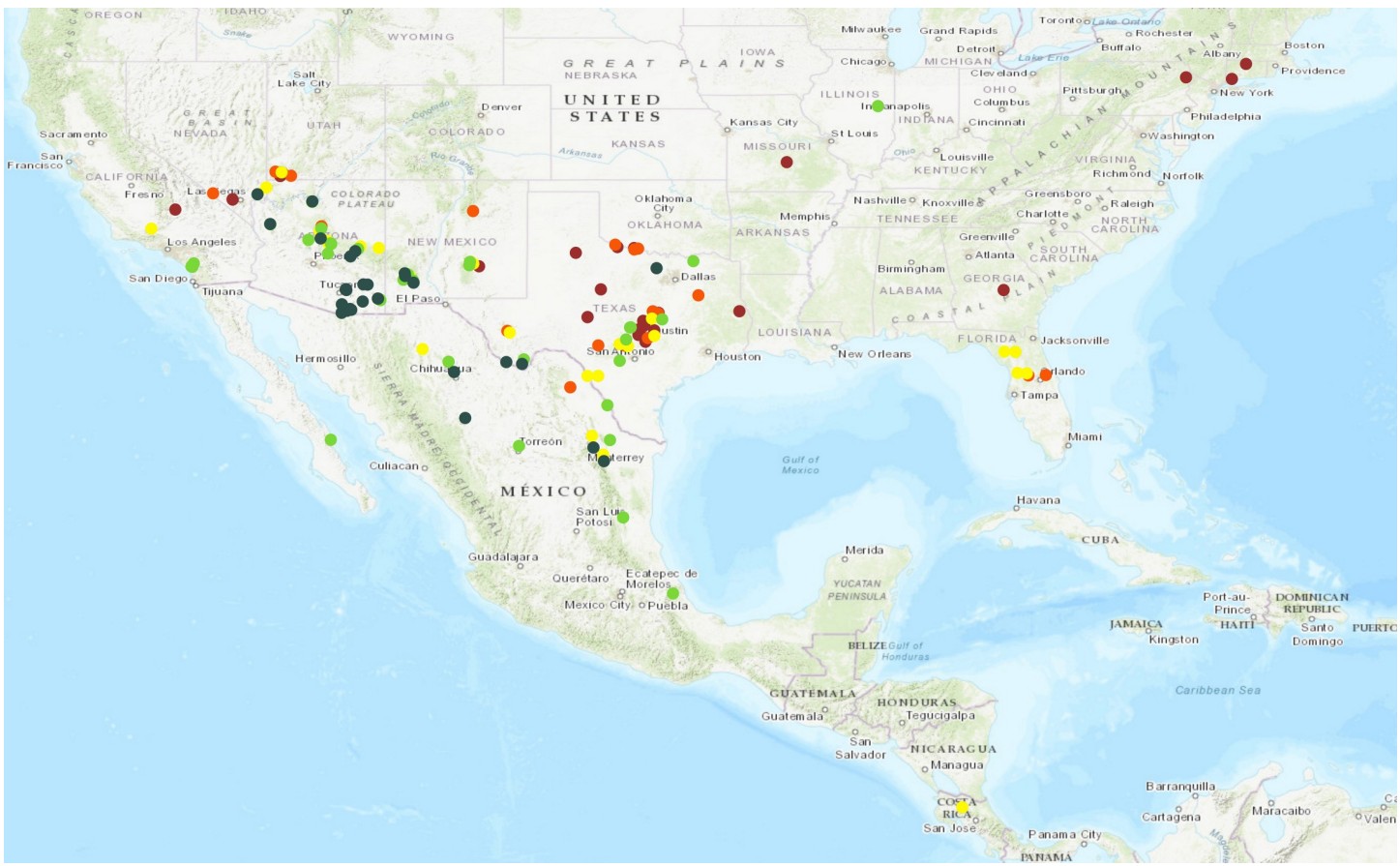

**Fig 1. Graphical presentation of the PD resistance of accessions from each collection location based on ELISA readings of stem tissue after greenhouse testing.** The data were grouped into five categories based on log-transformed estimations of bacterial levels: 1 = green, highly resistant (6.9–9.1); 2 = light green, resistant (9.2–11.1); 3 = yellow, moderately susceptible (11.2–13.1); 4 = orange, susceptible (13.2–15.1); and 5 = burgundy, highly susceptible (15.2 and above).

A group of 63 accessions had ELISA values in category 2, including five accessions with PD resistance on chr14. The CMI category values ranged from 1.0–4.0 with five accessions having LS/LL values of 4.0–5.0. Interestingly, the accession ANU46 collected from Arizona had high values for both CMI and LS/LL indicating that lower ELISA values do not necessarily mean that the plant will have reduced cane and leaf symptoms. Two accessions, ANU63 and T03-06S02, had good cane maturation but higher scores for LS/LL (S1 and S3 Tables), suggesting that both parameters of disease manifestation are independent of each other, and that there is genetic variation among different accessions for the ability to lignify the stem evenly and display leaf scorch symptoms. Twenty-seven accessions in this group were identified as *V. arizonica* based on morphological features. Almost all of the accessions in this group were collected from Mexico and bordering states with the exception of two *V. aestivalis* (DVIT1416, DVIT1609) accessions from Florida and Illinois (S1 and S3 Tables). Both accessions had moderate scores for CMI and LS/LL.

A group of 57 accessions had Category 3 ELISA results. Only 18 accessions in this group were identified as *V. arizonica* based on their morphology; the other accessions were within 12 other grape species (S1 and S3 Tables). The CMI results for these 57 accessions were in all five classes from very good cane maturation (1.0) to necrotic spots and multiple green islands (5.0). There were accessions with lower CMI and higher scores for LS/LL. Three accessions of

*V. shuttleworthii* (DVIT2387, DVIT1714, and DVIT2249.09) from Florida had intermediate bacteria levels, but high values for CMI and LS/LL. All of them had uneven cane maturation and leaf scorch symptoms under our greenhouse screening protocol of high temperatures and restricted irrigation. Similarly, one accession of *V. tiliifolia* collected from Costa Rica had intermediate ELISA results but high scores for CMI and LS/LL.

A group of 94 accessions had category 4 or 5 ELISA values (S1 and S3 Tables). This group contained accessions of *V. berlandieri*, *V. cinerea*, *V. aestivalis*, *V. girdiana* and *V. rupestris*. Twenty-two accessions of *V. berlandieri* were tested in the greenhouse and, with the exception of TX16-034, all had high ELISA scores. Only two accessions of *V. berlandieri* (TX *berlandieri* Male, and DVIT2220.04) had a moderate CMI index and low values for LS/LL. On the other hand, 28 tested accessions of *V. cinerea* spanned the full range of responses from resistant to very susceptible based on their ELISA, CMI and LS/LL values. Most of the resistant *V. cinerea* accessions were collected from Mexico (S1 and S3 Tables). A majority of the *V. aestivalis* accessions had high ELISA values, but their LS/LL values were moderate indicating their ability to tolerate higher bacterial levels without showing excessive leaf scorch. The 15 tested accessions of *V. girdiana* also displayed a wide range of responses from resistant to susceptible. Two accessions from this species were identified as having the *PdR1* locus in a previous study (S1 Table) [27]. All tested accessions of *V. rupestris* had high bacteria levels in the stem and high values for CMI and LS/LL, indicating susceptibility to PD.

The accessions identified as *V. arizonica* based on leaf morphology were the most resistant with low bacteria levels in the stem, and low CMI and LS/LL values. Among the 110 tested accessions collected from Mexico and bordering states, 82 had ELISA values in category 1 or 2, and with few exceptions, had very low CMI and LS/LL values (S1 and S3 Tables). Only nine accessions of *V. arizonica* had higher ELISA, CMI and LS/LL values, and most of these were collected from Utah.

Table 3 presents the greenhouse screening results from 12 reportedly resistant accessions released from southeastern US grape breeding programs. Their designation as PD resistant was based on survival in field trials and the lack of PD symptoms under natural infection. Two accessions (Blanc du Bois and Roucaneuf) were used as intermediate reference plants in all of this study's greenhouse experiments. With the exception of Florilush and Midsouth, all other accessions had high ELISA values (category 4 and 5) indicating that they can tolerate high levels of bacteria. However, they had variable CMI and LS/LL values; none were devoid of LS/LL symptoms and they had moderate to severe necrotic green islands on the stem. Only Florilush and Midsouth lignified normally with some leaf scorch and leaf loss. Blanc du Bois and Roucaneuf were similar to each other in terms of ELISA and LS/LL values, but their CMI values varied.

The three parameters that were used to determine PD resistance were also significantly different when comparing the southeastern and southwestern grape species (Table 4A). The LSM comparisons clearly separated the southwestern species, which had lower ELISA, CMI and LS/LL values (Table 4B).

## Genotyping of *Vitis* species

Initially, 346 accessions were genotyped with 24 nuclear SSR markers. A total of 20 accessions were excluded from further analysis due to missing data at 10 or more loci. Similarly, four markers were excluded due to difficulty in resolving single base pair variations or missing data, resulting in a set of 326 individuals genotyped at 20 loci. S2 Table details chromosome assignment, % missing data, number of alleles observed, and observed and expected heterozygosity for each locus for the 326 accessions. S4 Table provides the allelic data for the 326

**Table 3. List of 12 southeastern US varieties (with the exception of Roucaneuf) reported to be Pierce's disease (PD) resistant based on the survival rate and visual symptoms of leaf scorch under natural disease pressure in the field.** The two italicized and bolded accessions are reference plants, included in every greenhouse screening experiment. The other ten accessions were tested in earlier years of the PD resistance breeding program under greenhouse conditions and bacterial populations were quantified by ELISA. Cane maturation index (CMI) and leaf scorch/leaf loss symptoms were also recorded.

| Genotype | Reported PD resistant grape species in background and parents | No. of times tested | Mean CFU/ml (category) | Mean CMI (category) | Mean LS-LL (category) |
|---|---|---|---|---|---|
| Florilush | *V. aestivalis* ssp. *smalliana* and *V. champinii*, Dog Ridge × Tampa | 2 | 11.1 (2) | 0.2 (1) | 2.6 (3) |
| MidSouth | *V. champinii*, DeGrasset × Galibert 255–5 | 4 | 10.7 (2) | 0.2 (1) | 1.7 (2) |
| Stover | *V. shuttleworthii* and *V. lincecumii*, Mantey × Roucaneuf | 2 | 11.7 (3) | 1.5 (2) | 3.0 (3) |
| ***Blanc du Bois*** | *V. aestivalis* ssp. *smalliana*, D6-148 × Cardinal | 47 | 13.5 (4) | 3.1 (4) | 2.4 (3) |
| DC1-39 | *V. aestivalis* ssp. *smalliana*, W1521 × Aurelia | 3 | 13.4 (4) | 2.0 (2) | 3.2 (4) |
| MissBlue | *V. champinii*, Dog Ridge × Moore Early | 1 | 14.4 (4) | 4.7 (5) | 5.0 (5) |
| ***Roucaneuf*** | Complex French hybrid with multiple grape species in its pedigree, resistance possibly from *V. lincecumii*), Seibel 6468 × Seibel 6905 | 47 | 13.5 (4) | 1.4 (2) | 2.2 (3) |
| Suwannee | *V. aestivalis* ssp. *smalliana*, C5-50 × F8-35 | 2 | 14.1 (4) | 3.1 (3) | 2.6 (3) |
| Tamiami | *V. shuttleworthii*, Fennell 6 × Malaga (Dabouki) | 1 | 13.8 (4) | 3.8 (4) | 2.8 (3) |
| Tampa | *V. aestivalis* ssp. *smalliana*, Fla. 43–47 × Niagra | 3 | 13.7 (4) | 3.0 (3) | 3.0 (3) |
| Blue Lake | *V. aestivalis* ssp. *smalliana*, Fla. 43–47 × Caco | 2 | 15.0 (5) | 3.4 (4) | 3.7 (4) |
| Pixiola | *V. aestivalis*. spp. *simpsonii* | 2 | 15.6 (5) | 5.2 (5) | 3.8 (4) |

accessions. Overall, 20 markers represented 15 of the 19 grape chromosomes and only 2.2% of the data were missing. The number of alleles at each locus ranged from 10 (VVIq52) to 50 (VVIv67). The observed heterozygosity was lower than expected heterozygosity for all markers (S2 Table).

## Genetic diversity and clustering analysis

The genetic diversity of the 326 accessions was evaluated with a model-based clustering method implemented in the program STRUCTURE and by PCoA. The delta K value calculated from the output of STRUCTURE was 650 at K = 2 compared to less than 3 at all other K

**Table 4. Two-way ANOVA of the three groupings of three Pierce's disease evaluation parameters: ELISA readings of bacterial levels in the stem (CFU/ml), cane maturation index (CMI), and leaf scorch/leaf loss (LS/LL).** a. There was significant variation among the three groups for the three parameters. b. Least square means (LSM) comparisons of three Pierce's disease evaluation parameters based on ELISA readings of bacterial levels in the stem (CFU/ml), cane maturation index (CMI), and leaf scorch/leaf loss (LS/LL) using Tukey's test for three groups. Results significantly separate the southwestern US species from the southeastern US grape species across all three parameters of disease evaluation. Confidence levels (CL) were established at 95%.

| Disease evaluation parameter | No. of parameters | Degrees of Freedom | Sum of Squares | F Ratio | Prob > F |
|---|---|---|---|---|---|
| CFU/ml | 2 | 2 | 573.81 | 49.63 | < .0001* |
| CMI | 2 | 2 | 156.23 | 34.68 | < .0001* |
| LS/LL | 2 | 2 | 56.45 | 25.83 | < .0001* |

| Disease evaluation parameter | Population | LSM | Std. error | Lower CL | Upper CL | Group |
|---|---|---|---|---|---|---|
| CFU/ml | Southwestern US grape species | 10.59 | 0.21 | 10.19 | 11.00 | A |
|  | Southeastern US grape species | 13.43 | 0.27 | 12.91 | 13.96 | B |
|  | Admix of two groups | 14.18 | 0.45 | 13.30 | 15.06 | B |
| CMI | Southwestern US grape species | 1.56 | 0.13 | 1.31 | 1.81 | A |
|  | Southeastern US grape species | 2.90 | 0.17 | 2.57 | 3.23 | B |
|  | Admix of two groups | 3.65 | 0.28 | 3.10 | 4.19 | B |
| LS/LL | Southwestern US grape species | 2.40 | 0.09 | 2.22 | 2.57 | A |
|  | Southeastern US grape species | 3.31 | 0.12 | 3.08 | 3.54 | B |
|  | Admix of two groups | 3.48 | 0.19 | 3.10 | 3.86 | B |

values indicating division of genotypes into two groups (S5A Table). The Q-values (proportion of a given individual's genome that originated from a given population) assigned by STRUCTURE for 326 accessions in two groups are displayed in S6 Table. The threshold of 0.80 was selected to assign accessions to a particular group: a total of 140 accessions were assigned to group 1, 144 accessions in group 2 and 42 accessions were not fully assigned to either group. Fig 2 displays the collection locations of the 326 accessions with the fractional group assignments; bright cyan for group 1, burgundy for group 2. The two groups show a distinct east-west division. Group 1 consists of grape species that are native to the southeastern US and Texas with a genetic continuum extending to eastern Mexico and Central America. The species designation of accessions in group 1 were *V. aestivalis*, *V. labrusca*, *V. shuttleworthii*, *V. vulpina*, *V. champinii*, *V. biformis*, *V. nesbittiana*, *V. tiliifolia*, *V. candicans*, *V. cinerea*, and *V. berlandieri*. Group 2 primarily consists of *V. arizonica*, *V. girdiana*, *V. acerifolia*, *V. treleasei* and *V. rupestris*, from the southwestern US and Mexico. Interestingly, 42 accessions which were not fully assigned to either group were collected from eastern Mexico and the bordering state of Texas (S1 Table). Texas seems to be a major hybridization zone where the range of many grape species overlap. S7 Table shows the summary of F-statistic and gene flow within two main groups. For the southeastern group, the inbreeding coefficient ($F_{IS}$) per locus ranged from 0.036 (VVIb23) to 0.521(VVIq52), with an average of 0.211. The genetic differentiation ($F_{ST}$) of individual loci ranged from 0.031 (VMC7f2) to 0.275 (VVMD7), with an average value of 0.092, suggesting low genetic differentiation among the populations. Interestingly, the average gene flow (Nm) for southeastern group was 3.340, a much higher value than the southwestern group indicating more geneflow is prevalent, however, it does not increase the genetic differentiation. On the other hand, southwestern group had lower gene flow but higher genetic differentiation (S7 Table).

The hierarchal STRUCTURE analysis was carried out to identify diversity within each group. The threshold of 0.50 was used to assign accessions to develop two data sets for the second round of STRUCTURE. A range of genetic clusters (K) from 1 to 10 using the admixture model and 30 replications for each K were used for both runs and the threshold of 0.80 was selected to assign accessions to a particular group. The delta K values indicated four distinct groups within the southeastern species (S5B Table), and three distinct groups within the southwestern species (S5C Table). S6 Table presents the Q-values assigned by STRUCTURE for the first and second-round of analysis. Fig 3A shows the bar plot of first and second round of the STRUCTURE results. The four southeastern species groups consisted of *V. cinerea/berlandieri*, *V. candicans/monticola*, *V. aestivalis/labrusca* and *V. cinerea/tiliifolia* collected from Mexico. Nineteen samples were admixture and 12 of them were also shown to be admix in the first round of analysis. The southwestern group divided into *V. arizonica*, *V girdiana* with *V. arizonica* accessions collected from the Big Bend area of Texas and *V. arizonica* collected from Mexico with 28 admix samples that were not assigned to any group.

Principal coordinate analysis explained 12.38% of the variation among 326 accessions on two axes (S4 Fig). The color coding of the STRUCTURE assignment was used for the PCoA display. The PCoA analysis was also carried out on each group and results were consistent with the Bayesian clustering analysis (Fig 3B). Four distinct groups were identified within southeastern grape species that explained 14.50% variation with admix genotypes in between the groups. Similarly, the southwestern group divided into three sub-groups that explained 15.77% of the variation. Accessions that were not fully assigned to a structure group were positioned between the clusters in the PCoA analysis. The accessions collected from the Big Bend region (group 2.2), and from Mexico (group 2.3) were called *V. arizonica* based on their morphological features. It is most likely that they are complex hybrids of other species that are potentially not represented in this study set.

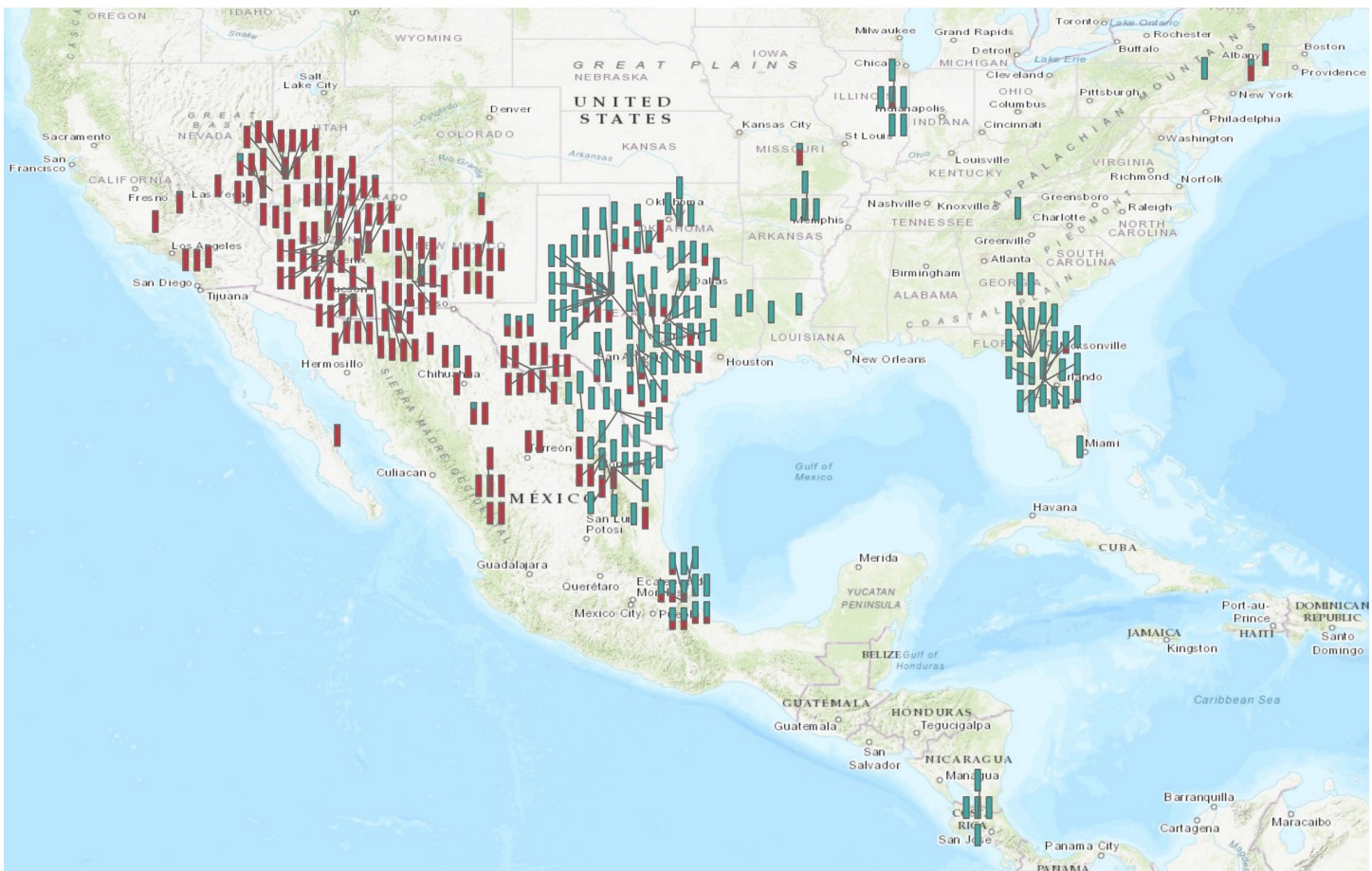

**Fig 2. Graphical presentation of the phylogeographic results based on the membership estimates generated by the STRUCTURE program for each genotype.** Bright cyan represents group 1 consisting of the southeastern US grape species. Moderate red represents group 2 consisting of the southwestern US grape species. Details of the accessions in each group are provided in supplementary S1 Table.

Table 5 presents the summary of PD evaluation results across groups identified in the second round of the STRUCTURE analysis. Within the southeastern group, *V. cinerea* accessions collected from Mexico and *V. candicans* were the only species with strong resistance to PD. The accessions of *V. cinerea* collected from Texas were genetically distinct and susceptible to the disease (Table 5, Fig 3B). Similarly, accessions of *V. aestivalis* and *V. labrusca* were also susceptible to the disease when tested under these greenhouse conditions. In the southwestern group, accessions of *V. arizonica* appeared in all three clusters and were highly resistant to PD. They were collected from different geographic regions. The accessions of *V. girdiana* and *V. treleasei* showed moderate resistance to the disease.

## Discussion

In this study, we surveyed 326 grape accessions of 19 grape species with molecular markers and combined population genetic diversity information with the results of greenhouse-based PD resistance evaluations to determine the range of PD resistance in wild grape species. Historic breeding lines from the southeastern US, reported to be resistant to the disease, were also tested. Pierce's Disease resistance status was quantified by measuring the bacterial levels in the

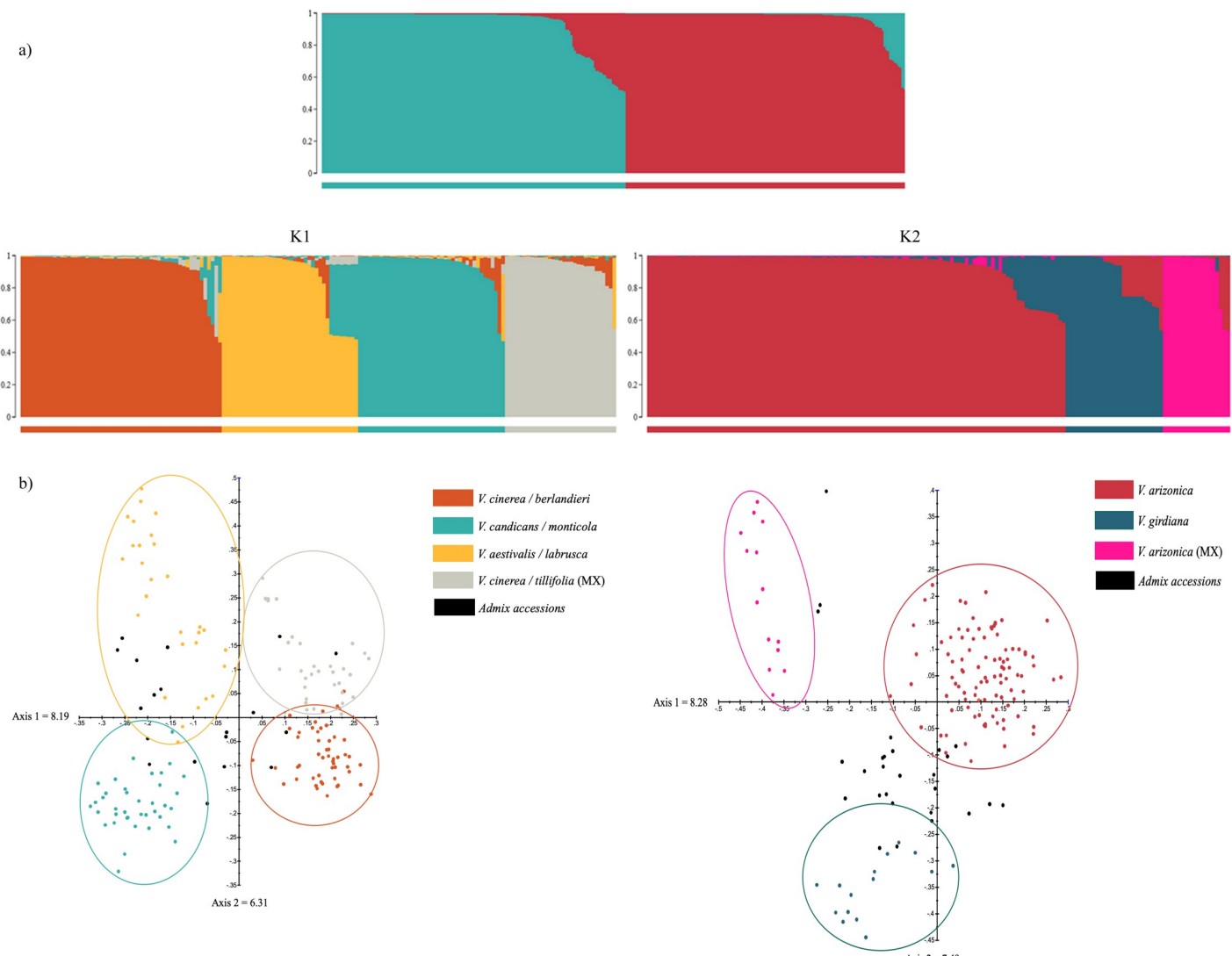

**Fig 3.** a) Bayesian STRUCTURE bar plot of membership for 326 accessions (K = 2). Bright cyan represents group 1 (K1 = southeastern US grape species) and moderate red represents group 2 (K2 = southwestern US grape species). STRUCTURE analysis on each group resulted in four distinct sub-groups within K1 and three sub-groups within K2. For details of membership coefficient values see S6 Table. b) Principal coordinate analysis for each group resulted in the comparable results to the Bayesian analysis. The color coding of the STRUCTURE assignment was used for the PCoA display.

stem and recording cane maturation, leaf scorch and leaf loss symptoms. These genetic diversity and PD evaluation data revealed major trends.

## Genetic divide between eastern and western US grape species and the presence of hybrid zones

Two distinct genetic groups were identified with a Bayesian clustering approach based on the allele frequencies of SSR markers (Figs 1 and 3, S5 and S6 Tables). Group 1 consisted of grape species native to the southeastern US, eastern Mexico and Costa Rica, and group 2 primarily consisted of grape species accessions from the southwestern US and Mexico. Principal coordinate analysis also revealed two main groups of species (S4 Fig). Geographically, the two groups were separated by the Sierra Madre Oriental mountain range, which extends along the eastern

**Table 5. Summary of PD resistance across different species groups as defined by the hierarchal STRUCTURE analysis.** For the three parameters, lower scores exhibit higher resistance to the disease.

| | | Main species | No. of accessions | CFU/ml | CMI | LS/LL | No. with no data |
|---|---|---|---|---|---|---|---|
| Group 1 (Southeastern US, MX, CR) | Group 1.1 | *V. berlandieri* | 17 | 15.7 | 3.2 | 3.7 | 2 |
| | | *V. cinerea* (US) | 30 | 15.8 | 3.4 | 3.4 | 23 |
| | | Other | 4 | 13.0 | 3.7 | 3.9 | 1 |
| | Group 1.2 | *V. candicans* | 10 | 10.4 | 1.7 | 2.6 | 0 |
| | | *V. monticola* | 6 | 14.0 | 3.0 | 2.7 | 1 |
| | | Other | 13 | 13.9 | 3.5 | 3.7 | 3 |
| | Group 1.3 | *V. aestivalis* | 29 | 14.3 | 3.7 | 3.0 | 18 |
| | | *V. labrusca* | 4 | 15.5 | 4.6 | 4.0 | 2 |
| | | Other | 5 | 12.5 | 3.0 | 3.2 | 1 |
| | Group 1.4 | *V. cinerea* (MX) | 20 | 11.9 | 1.8 | 3.2 | 0 |
| | | *V. tiliifolia* | 7 | 14.0 | 5.3 | 4.6 | 4 |
| | | Other | 6 | - | - | - | 6 |
| | ADMIX | | 19 | 13.4 | 3.6 | 3.0 | 11 |
| | Total | | 170 | | | | 72 |
| Group 2 (Southwestern US and MX) | Group 2.1 | *V. arizonica* (US) | 79 | 9.8 | 1.1 | 2.1 | 0 |
| | | *V. treleasei* | 17 | 12.0 | 1.7 | 2.9 | 1 |
| | | Other | 1 | 15.8 | 3.5 | 4.1 | 0 |
| | Group 2.2 | *V. girdiana* | 11 | 12.1 | 3.2 | 2.9 | 0 |
| | | *V. arizonica* (Big Bend region) | 16 | 11.0 | 1.7 | 2.1 | 6 |
| | | Other | 0 | - | - | - | 0 |
| | Group 2.3 | *V. arizonica* (MX) | 8 | 9.3 | 1.6 | 2.2 | 0 |
| | | *V. rupestris* | 7 | 15.4 | 4.8 | 4.3 | 0 |
| | | Other | 3 | 16.1 | 3.0 | 4.0 | 0 |
| | ADMIX | | 14 | 12.2 | 2.4 | 3.1 | 0 |
| | Total | | 156 | | | | 7 |

side of Mexico. This east/west dividing line extends north through central Texas and separates the drier western US from the wetter eastern US. The current topology of North America, resulting from geologic events over millennia, gave rise to the current coastal and central plains, extensive mountain ranges that stretch thousands of miles (the Rocky Mountains, Sierra Madre) and desert regions (Sonora, Chihuahua) [47]. This topology also created unique niches for different grape species kept separate by phenological differences in flowering dates, and water availability [30,48,49]. The results of hierarchal STRUCTURE and PCoA analysis on each group were comparable and further split each cluster into distinct species groups with overlapping habitats (Fig 3A and 3B). We identified four sub-clusters within the southeastern grape species, V. *cinerea*/*berlandieri*, V. *candicans*/*monticola*, V. *aestivalis*/*labrusca*, and *V. cinerea*/*tiliifolia* collected from Mexico and Costa Rica. These species are reported to be phylogenetically close to each other and also exhibit greater overlap of habitat [30,50,51]. However, this is the first time we have seen differentiation among accessions of *V. cinerea* that were collected from different areas indicating regional types or varieties. The accessions of *V. cinerea* collected from Mexico represent a very unique gene pool.

Within southwestern grape species, three sub-clusters were identified and accessions of *V. arizonica* were present in all of them indicating a higher level of variability than has been reported earlier [51]. *Vitis arizonica* hybridizes without difficulty with *V. girdiana* in its western range, with *V. riparia* in its northeastern range, and *V. acerifolia*, *V. candicans* and *V. cinerea* in its eastern range. This hybridization makes it difficult to determine where species

boundaries exist, to what extent intraspecies variation occurs, and whether these sympatric species are giving rise to new species. It can be very difficult to distinguish many of these hybrid forms based on morphological features alone. A thorough taxonomic and genetic analysis of the southwestern US *Vitis* species and the Mexican *Vitis* is required to gain a better understanding of this important germplasm.

The overall east-west genetic divide of grape species that we have observed in this study has also been identified in other plant and animal taxa. Escalante et al. [52] analyzed 40 Mexican plant and animal taxa whose range extends to both the Nearctic and Neotropical regions, and also identified two main clades with an east-west pattern. The plant and animal taxa from the Mexican Gulf, from Tamaulipas to Yucatan, were in one clade, which forms the lowland region of eastern Mexico along the Caribbean coastline and as far north as the southern US. The other clade included biota from central and western Mexico [52]. We identified a similar distribution pattern for grape species. It is interesting to note that the climatic conditions for the species in group 1 are more tropical with more precipitation, while Group 2 consisted of grape species from more arid climates (Fig 2). Results from this study indicate that the Sierra Madre Oriental (to the east and often considered an extension of the southern Rocky Mountains) is a major physical barrier that has kept Mexican grape species apart for many thousands of years. The Sierra Madre Occidental (to the west), along with the Sierra Madre Oriental, enclose the Mexican plateau that merges with the Basin and Range provinces of the southwestern US. This region has an extraordinary topography with numerous small mountain ranges that act as rain collectors and are separated by the desert plains of Sonora and Chihuahua. It is not surprising to find that mountain ranges and dry desert regions act as natural barriers for gene flow. We also identified lower levels of gene flow among southwestern grape species in comparison to the grape species from the southeast that have higher levels of habitat overlap and more genetic continuum (S7 Table).

Migrating birds and their berry feeding are also a primary factor in the dispersal of grape seeds over long distances, helping to expand the range grape species. Four major bird flyways or migratory routes exist in North America (https://www.fws.gov/refuge/arctic/birdmig.html). The central flyway route covers Texas and Arizona where many *Vitis* species exist and where we detected several accessions that were hybrids of two or more grape species. This is also the region where the ranges of different grape species overlap providing further opportunities for hybridization and development of new variant forms capable of adapting to climatic niches.

## Geographic pattern of resistance to PD in grape species

A total of 266 accessions were evaluated for PD resistance using an established greenhouse screening protocol (Table 5, S1 and S3 Tables). The presentation of PD screening results on the map demonstrates an east-west axis with stronger resistance to PD present in western grape accessions in terms of lower *X. fastidiosa* levels in the stems, better cane lignification, and reduced leaf loss and leaf scorch (Fig 1, S1 and S3 Tables). Most accessions of southeastern grape species (with exception of *V. candicans*, and *V. cinerea* that was collected from Mexico) had higher levels of *X. fastidiosa* in the stems and more severe symptoms in the stems and leaves, all of which were intensified under high temperature greenhouse screen conditions.

A similar trend was observed in PD resistant breeding lines from the southeast US, which had higher levels of *X. fastidiosa* and acute symptoms on the stem and leaves (Table 3). These breeding lines in their native habitat can have mild symptoms and be long-lived in the field [4,14–16]. A possible explanation for this disparity is that southeastern US grape species are not resistant to PD, but instead are tolerant and the stem and leaf symptoms of PD are suppressed by wetter, more humid conditions. After infection, *X. fastidiosa* inhabits and spreads

within the xylem. The infection initiates a plant response that results in vascular occlusions, predominantly tyloses, to limit bacterial spread. This response also decreases water transport up to 90% in susceptible plants [53]. Warm humid climates promote more vegetative growth, which may allow infected plants to dilute the infection and tolerate blocked vessels for a longer time.

In general, PD symptom development is highly correlated with the number of clogged vessels [54] and with higher levels of bacteria in the stem [36]. In this study, ELISA values had 76% correlation to CMI and 68% to LS/LL. However, we found many accessions from different species that had high ELISA values but lower values for the CMI and LS/LL and vice versa (S1 and S3 Tables). These results indicate that there are differences in the manifestation of PD symptoms among different grape species and more research is needed to understand *X. fastidiosa*'s pathogenicity, particularly in grape species endemic to regions with high rainfall during summer months. *Vitis candicans* collected from Texas and *V. cinerea* accessions collected from Mexico were the exception in the southeastern grape species group possessing accessions with PD resistance in the greenhouse screen. *Vitis candicans* is known to have preference for warm humid conditions [51] and is found abundantly with vigorous growth in central to eastern Texas [30]. In this study we tested twelve accessions of *V. candicans* that were collected during field collection trips in late nineties and early twenties. All of them were promising with strong PD resistance. Future studies should focus on this valuable source of resistance to PD and other grape pests.

Among the southwestern grape species, *V. treleasei* (a glabrous form of *V. arizonica*) [30], and *V. girdiana* showed moderate PD resistance, while the strongest PD resistance was found in accessions of *V. arizonica*–which had 82 out of 110 accessions with low bacterial levels and low CMI and LS/LL values (Table 5, S1 and S3 Tables). Nine accessions with the *PdR1* locus, identified in earlier studies, were also pure forms of *V. arizonica* or apparent hybrids with this species that ranges across the arid southwestern US and northern and northwestern Mexico [27,37,55]. Results from this study shows that the accessions of *V. arizonica* collected from different geographical regions of the southwest US and Mexico belonged to three distinct genetic subgroups and all of them were highly resistant to PD. In contrast, accessions of *V. cinerea* had one genetic subgroup resistant to the disease and the other was susceptible (Table 5). Forms of *V. arizonica* are morphologically adapted to droughty and xeric conditions, but they are most often found in wet areas within these isolated xeric regions such as springs, creeks, and catchment basins. It hybridizes with other grape species when their ranges overlap or are connected by migratory flight patterns of birds. It is important to collect more germplasm from Mexico and Central America to carry out phylogenetic analysis as well as disease evaluations with this germplasm to confirm the results of this study, and gain more insight into the extent and spread of PD resistance in different geographic regions of Mexico and Central America.

## Conclusions

Grape species from Mexico and southwestern US do not exist on a genetic continuum–the Sierra Madre Oriental mountain range acts as a phylogeographic barrier to the south and to the north central Texas and the Great Plains separate wet from arid climates of the US. *Vitis candicans* and Mexican *V. cinerea* accessions were the only grape species with strong PD resistance in the southeastern group. Among southwestern grape species, *Vitis arizonica* accessions displayed strong PD resistance despite genetic variation and *V. girdiana* and *V. treleasei* showed moderate resistance.

## Supporting information

**S1 Fig. Seven reference plants at 12 weeks post inoculation.** The reference plants were included in all greenhouse screening experiments to compare screen results across different

experiments and years.
(TIF)

**S2 Fig. ELISA values of seven reference plants [two resistant (b43-17, U0505-01), three intermediate (U0505-35, Roucaneuf, Blanc du Bois), and two susceptible (U0505-22, Chardonnay)] across 19 screens.**
(TIF)

**S3 Fig.** a) Cane maturation index; b) Leaf scorch/leaf loss. For both a and b, five categories were used: 1 (0–1), 2 (1.1–2.0), 3 (2.1–3.0), 4 (3.1–4), and 5 (4.1and above). The color scheme employed in Fig 3 was used for the five categories (dark green = 1, light green = 2, yellow = 3, orange = 4, and burgundy = 5).
(TIF)

**S4 Fig. Principal Coordinates Analysis constructed with genotypic data from 20 SSR markers on 326 accessions using DARWIN software.** Axis 1 and 2 represent 8.49 and 3.89 percent of the variation, respectively. The color coding of STRUCTURE assignment was used for the PCoA display (bright cyan = group 1, moderate red = group 2, and black = admixture).
(TIF)

**S1 Table. List of 346 accessions and their species designation based on the morphology.** A total of 20 accessions were excluded from the genetic analysis due to lack of data with more than four markers. Greenhouse screening for Pierce's disease was completed for 266 accessions. Least square means were calculated for the bacterial count (colony forming unit—CFU), cane maturation index (CMI) and leaf loss/leaf scorch (LS/LL). Thirteen bold and underlined accessions were found to have PD resistance on chromosome 14 in previous studies.
(XLSX)

**S2 Table. List of 24 SSR markers with chromosome designation, and fluorescent label.** The % missing data, number of observed alleles (Na), observed heterozygosity (Ho), and expected heterozygosity (He) were determined with 326 accessions that were included in the final analysis.
(XLSX)

**S3 Table. List of 266 accessions with species designation and collection location that were tested for PD resistance under greenhouse conditions.** Least square mean values for cfu/ml, CMI index and LS/LL from Table S1 were divided into 5 categories; 1 = 6.9–9.1, 2 = 9.2–11.1, 3 = 11.2–13.1, 4 = 13.2–15.1, 5 = above 15.2. The five categories for the CMI and LS/LL scores were: 1 (0–1), 2 (1.1–2.0), 3 (2.1–3.0), 4 (3.1–4), and 5 (4.1and above). Accessions are organized from lower to higher CMI values. Underlined and bold accessions were identified to carry *PdR1* resistance locus.
(XLSX)

**S4 Table. Simple sequence repeat marker data for 326 accessions that were included for the genetic analysis.** ND is no data.
(XLSX)

**S5 Table.** Delta K value as a function of K based on 20 runs for the first round of STRUCTURE analysis indicates two genetic clusters (a) of southeastern and southwestern grape species. STRUCTURE analysis was run on each group with 10 runs and 30 replications. Delta K value for the southeastern group indicates four genetic groups within 170 accessions (b) and three genetic groups within 156 accessions of southwestern origin (c).
(XLSX)

**S6 Table. Population assignment to two groups and Q-values were determined with the STRUCTURE program.** Hierarchal STRUCTURE analysis on each group clearly divided each group into clusters of distinct species. K1 cluster divided into four groups and K2 cluster into three distinct groups.
(XLSX)

**S7 Table. Summary of F statistics and gene flow for the 20 loci over all populations within two groups.**
(XLSX)

## Author Contributions

**Conceptualization:** Summaira Riaz, Claire C. Heinitz, Karla G. Huerta-Acosta, M. Andrew Walker.

**Data curation:** Summaira Riaz, Alan C. Tenscher, M. Andrew Walker.

**Formal analysis:** Summaira Riaz, Claire C. Heinitz, M. Andrew Walker.

**Funding acquisition:** M. Andrew Walker.

**Investigation:** Summaira Riaz, Alan C. Tenscher, Claire C. Heinitz, Karla G. Huerta-Acosta, M. Andrew Walker.

**Methodology:** Summaira Riaz, Claire C. Heinitz, Karla G. Huerta-Acosta.

**Resources:** M. Andrew Walker.

**Supervision:** Summaira Riaz, Alan C. Tenscher, M. Andrew Walker.

**Writing – original draft:** Summaira Riaz, Claire C. Heinitz.

**Writing – review & editing:** M. Andrew Walker.

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
