## [Decision Letter · Decision Letter 0]

24 Sep 2020

PONE-D-20-26579

Genetic analysis reveals an east-west divide within North American Vitis species that mirrors their resistance to Pierce's disease.

PLOS ONE

Dear Dr. Walker,

Thank you for submitting your manuscript to PLOS ONE. After careful consideration, we feel that it has merit but does not fully meet PLOS ONE’s publication criteria as it currently stands. Therefore, we invite you to submit a revised version of the manuscript that addresses the points raised during the review process.

We look forward to receiving your revised manuscript.

Kind regards,

Tzen-Yuh Chiang

Academic Editor

PLOS ONE

Journal Requirements:

2. We note that Figures 1, 2 and S3 in your submission contain map images which may be copyrighted. All PLOS content is published under the Creative Commons Attribution License (CC BY 4.0), which means that the manuscript, images, and Supporting Information files will be freely available online, and any third party is permitted to access, download, copy, distribute, and use these materials in any way, even commercially, with proper attribution. For these reasons, we cannot publish previously copyrighted maps or satellite images created using proprietary data, such as Google software (Google Maps, Street View, and Earth). For more information, see our copyright guidelines: http://journals.plos.org/plosone/s/licenses-and-copyright.

2.1.    You may seek permission from the original copyright holder of Figures 1, 2 and S3 to publish the content specifically under the CC BY 4.0 license. 

2.2.    If you are unable to obtain permission from the original copyright holder to publish these figures under the CC BY 4.0 license or if the copyright holder’s requirements are incompatible with the CC BY 4.0 license, please either i) remove the figure or ii) supply a replacement figure that complies with the CC BY 4.0 license. Please check copyright information on all replacement figures and update the figure caption with source information. If applicable, please specify in the figure caption text when a figure is similar but not identical to the original image and is therefore for illustrative purposes only.

3. Please upload a copy of Supporting Information Tables 1-7 which you refer to in your text.

Reviewers' comments:

Reviewer's Responses to Questions

**Comments to the Author**

1. Is the manuscript technically sound, and do the data support the conclusions?

Reviewer #1: Yes

2. Has the statistical analysis been performed appropriately and rigorously? 

Reviewer #1: Yes

3. Have the authors made all data underlying the findings in their manuscript fully available?

Reviewer #1: Yes

4. Is the manuscript presented in an intelligible fashion and written in standard English?

Reviewer #1: Yes

5. Review Comments to the Author

Reviewer #1: This study used an extensive sampling of Vitis species and populations from the whole North America continent to study the phylogenetic relationships of grape germplasm and their correlation with pierce’s disease resistance. This study paid much attention and works to evaluate disease resistance by quantifying the level of bacteria in stems and measuring Pierce’s disease symptoms on the canes and leaves. Results of 20 SSR markers to genotype 326 accessions of grape species well suggested a geographical divide in the southwest and southeastern US and Mexico. Two hundred sixty-six of these accessions, and an additional 12 PD resistant hybrid cultivars developed from southeastern US grape species, were evaluated for PD resistance. This work provide clear phylogenetic pattern for the grape genus in North America with interesting pierce’s disease resistance evaluation. But regretfully, no analyses were performed to test the correlation between this disease resistance trait and their phylogenetic and biogeographic relationships or explanation for the disease resistance pattern in North American grapes. Also, Results are too long and Discussion looks a little thin. Anyway, this study revealed an interesting phylogenetic pattern and the PD resistance pattern in North American grapes, and it is worthy to be accepted by the journal with some improvements

1. Phylogenetic history within Vitis is an interesting puzzle and topological relationships for the whole Vitis group including North America have been studied many times before. All these works should be compared and discussed in this study. Further, it is better to present a topological tree to show their phylogenetic relationships.

2. PD resistance data seems not to be well match phylogenetic divide of North American grapes, there is no clear east-west divide as shown in phylogenetic data. A correlation analyses should be used. Climate is a possible explanation for their different disease resistance, but there must have many other causes, such as from an evolutionary view.

3. Sample information including locations or sources for all the 346 accessions should be given in table or as a supplement.

4. To have a better presentations, subgroups in the two main groups should be also marked in Figure 2. If possible, hybridization zone and the Sierra Madre Oriental should also be noted on Figure 2 since they have been mentioned many times in the discussion.

5. Line 52, Muscadinia rotundifolia  Vitis rotundifolia

6. PLOS authors have the option to publish the peer review history of their article (what does this mean?). If published, this will include your full peer review and any attached files.

Reviewer #1: No

---

## [Author Response · Author response to Decision Letter 0]

30 Oct 2020

All my comments are covered in the uploaded Response to Reviewers.

---

## [Decision Letter · Decision Letter 1]

23 Nov 2020

Genetic analysis reveals an east-west divide within North American Vitis species that mirrors their resistance to Pierce's disease.

PONE-D-20-26579R1

Dear Dr. Walker,

We’re pleased to inform you that your manuscript has been judged scientifically suitable for publication and will be formally accepted for publication once it meets all outstanding technical requirements.

Kind regards,

Tzen-Yuh Chiang

Academic Editor

PLOS ONE

Additional Editor Comments (optional):

Reviewers' comments:

Reviewer's Responses to Questions

**Comments to the Author**

1. If the authors have adequately addressed your comments raised in a previous round of review and you feel that this manuscript is now acceptable for publication, you may indicate that here to bypass the “Comments to the Author” section, enter your conflict of interest statement in the “Confidential to Editor” section, and submit your "Accept" recommendation.

Reviewer #1: All comments have been addressed

2. Is the manuscript technically sound, and do the data support the conclusions?

Reviewer #1: Yes

3. Has the statistical analysis been performed appropriately and rigorously? 

Reviewer #1: Yes

4. Have the authors made all data underlying the findings in their manuscript fully available?

Reviewer #1: Yes

5. Is the manuscript presented in an intelligible fashion and written in standard English?

Reviewer #1: Yes

6. Review Comments to the Author

Reviewer #1: This manuscript has been largely and concisely revised though some minor comments were not accepted by the authors.

7. PLOS authors have the option to publish the peer review history of their article (what does this mean?). If published, this will include your full peer review and any attached files.

Reviewer #1: No

---

## [Editor Report · Acceptance letter]

10 Dec 2020

PONE-D-20-26579R1 

Genetic analysis reveals an east-west divide within North American *Vitis* species that mirrors their resistance to Pierce’s disease 

Dear Dr. Walker:

I'm pleased to inform you that your manuscript has been deemed suitable for publication in PLOS ONE. Congratulations! Your manuscript is now with our production department. 

Kind regards, 

on behalf of

Dr. Tzen-Yuh Chiang 

Academic Editor

PLOS ONE